# Division-of-Thoughts: Harnessing Hybrid Language Model Synergy for Efficient On-Device LLM Agents

## Abstract

With the emergence of edge AI application scenarios such as on-device web search assistants, enhancing the reasoning performance of large language models (LLMs) on edge devices has become an increasingly important topic. Due to the memory and computation limitations of edge devices, edge-cloud collaboration presents a promising solution, which involves deploying smaller LLMs locally while invoking larger-scale LLMs in the cloud. However, how to coordinate these two to balance cost and performance is a challenge. We propose a new collaborative reasoning framework called Division-of-Thoughts (**DoT**) to fully harness the synergy between locally deployed SLMs and cloud-based LLMs. DoT leverages a *Task Decomposer* to elicit the inherent planning abilities in language models to decompose user queries into smaller sub-tasks. We also employ a *Task Scheduler* to analyze the pair-wise dependency of sub-tasks and create a dependency graph, facilitating parallel reasoning of sub-tasks and the identification of key steps. To allocate the appropriate model based on the difficulty of sub-tasks, DoT leverages a *Plug-and-Play Adapter*, which is an additional task head attached to the SLM that does not alter the SLM's parameters. To boost the allocation of the adapter, We also design a self-reinforced tree search algorithm to create a high-qualiy sub-task allocation dataset. Extensive experiments on various benchmarks demonstrate that our DoT significantly reduces LLM costs while maintaining reasoning accuracy. Comparable to the best baseline methods, we reduce the average reasoning time and API costs by 66.12% and 83.57%, respectively. Our code can be accessed via the following link: https://anonymous.4open.science/status/DoT-F17C

## Keywords

LLM Agent, Personal Assistant, Edge-Cloud Collaboration

**ACM Reference Format:**
Anonymous Author(s). 2018. Division-of-Thoughts: Harnessing Hybrid Language Model Synergy for Efficient On-Device LLM Agents. In *Proceedings of Make sure to enter the correct conference title from your rights confirmation emai (Conference acronym 'XX).* ACM, New York, NY, USA, 13 pages. https://doi.org/XXXXXXX.XXXXXXX

## 1 Introduction

As web content continues to grow exponentially, on-device AI assistants have become essential tools for helping users navigate the increasingly complex online landscape. This trend has led to the widespread adoption of personal assistants such as Google Assistant, Apple Siri, Amazon Alexa, Alibaba Tmall Genie, and Xiaomi Xiao AI [13, 23], which have demonstrated their effectiveness in helping users digest enormous web content for tasks like web browsing [16], content searches [28], online shopping [25], and travel planning [5]. These AI-powered agents enable web applications to harness the rapid advancements in AI technology, delivering a more personalized and convenient user experience. Amid recent AI breakthroughs in Large Language Models (LLMs), the emergent capabilities of commonsense reasoning [32] and in-context learning [6] are widely regarded as a key component for the next generation of on-device agents [9]. Therefore, revolutionizing AI personal assistants with LLM agents has become an important research problem and a critical focus for applications [19].

However, deploying LLM agents on local devices presents significant challenges, as it is impractical to run LLMs with trillions of parameters on resource-constrained devices such as smartphones and personal computers [34]. Conversely, relying solely on cloud-based commercial LLMs raises concerns over privacy risks, unreliable connections, and high monetary costs [18, 19]. Recent research has focused on training smaller-scale language models and developing model compression techniques [20, 21, 33], with the goal of creating sub-10B parameter models that can be practically deployed on local devices, such as Llama 3 series [8] and Phi 3 series [1]. However, this approach introduces additional computational costs for training or compressing these models and inevitably results in performance degradation compared to full-size LLMs. Preliminary efforts have also explored the potential of edge-cloud collaboration, where tasks exceeding the capabilities of locally deployed Smaller-scale Sanguage Models (SLMs) are rerouted to more powerful cloud-based LLMs [3, 10]. Despite this, the significant performance gap between SLMs and LLMs often leads to a suboptimal trade-off between reasoning capabilities and cost.

To address this problem, our work draws inspiration from the fundamental economic concept of "division of labour," which posits that breaking down complex tasks into finer components often leads to more efficient solutions by allowing collaborative partners to fully exploit their respective strengths. Building on this idea, we propose a novel Division-of-Thoughts (**DoT**) framework to fully harness the synergy between locally deployed SLMs and cloud-based LLMs through sophisticated task decomposition and optimized sub-tasks allocation. Specifically, DoT leverages a *Task Decomposer*, a meta-prompt that combines "chain-of-thought"-like prompting [32] with carefully curated task decomposition examples. This approach taps into the inherent planning abilities of language models [29], enabling them to decompose user queries into smaller sub-tasks. Our core insight is that even complex user queries often contain a significant portion of simple sub-tasks that can be adequately handled by SLMs. Therefore, decomposing user

queries can lead to optimized collaboration on sub-task level, *i.e.*, unleashing the potential of "Division-of-Thoughts." Besides, DoT employs a *Task Scheduler* that analyzes the pair-wise dependency between sub-tasks and creates a dependency graph. This facilitates efficient and accurate scheduling by identifying parallel sub-tasks and assessing the structural importance of each task. More importantly, we design a self-reinforced training method to boost the task allocation capability of SLM, requiring no human annotation, but only the feedback of task execution. Our training method leverages a novel tree search algorithm that accounts for both the uncertainty of language models and the task performance to optimize sub-task allocation decisions. As a result, we can create a high-quality sub-task allocation dataset without human supervision. Based on which, we train a light-weight, plug-and-play adapter that significantly boosts SLM's ability to allocate sub-tasks. It is important to note that the adapter fully preserves the general capabilities of the SLM, as it does not modify SLM's parameters. Instead, it introduces a detachable decoding head specialized for sub-task allocation.

We conducted extensive experiments on six widely adopted LLM agent benchmarks, covering a variety of scenarios, including logical reasoning, web browsing, solving math problems, and common-sense reasoning. The results demonstrate that our DoT framework significantly reduces LLM costs while maintaining reasoning accuracy comparable to the best baseline methods across all benchmarks. Specifically, the average reasoning time and API costs are reduced by 66.12% and 83.57%, respectively, compared to the most accurate baseline methods. Besides, an ablation study confirms the effectiveness of our key model design choices. Furthermore, the DoT framework consistently achieves superior cost-accuracy trade-offs compared to task referral strategies that do not incorporate task decomposition across all budget settings, where the performance gain are particularly large at low cost setting.

To summarize, our contributions are three-fold:

- we present a novel Division-of-Thoughts (DoT) framework that fully exploits the synergy between locally deployed SLM and cloud-based LLM to power on-device agents with cost-effective LLM reasoning.
- We design a *Task Scheduler* to extract the dependency graph among sub-tasks, facilitating optimal sub-task scheduling.
- We propose a self-reinforced training method to boost SLM's task allocation accuracy without additional human annotation, and also preserve the general reasoning capabilities of SLM with detachable, plug-and-play adapter.

## 2 Related Work

### 2.1 Emergence and Extension of LLM Reasoning Capabilities.

In recent years, large language models (LLMs) have undergone remarkable and revolutionary advancements. As the scale of these models increases, LLMs exhibit emergent reasoning capabilities that are notably powerful [31]. Building upon these advancements, various prompt engineering techniques have further extended the reasoning capabilities of LLMs. Chain-of-Thought (CoT) [32] improves reasoning performance by incorporating manually crafted decomposition steps into the prompt, allowing the LLM to follow the step-by-step resolution process. Building on this, Zero-shot

CoT [15] achieves similar effects by simply adding the phrase "Let's think step by step," enabling LLMs to automatically decompose and execute tasks. From the linear structure of CoT, more complex frameworks have been introduced, such as Tree-of-Thought [36] and Graph-of-Thought [2], which expand LLMs' reasoning capabilities through branching reasoning paths. However, these methods face increased resource demands and time complexity, and they heavily rely on manually predefined steps tailored to specific tasks, which significantly limits their generalizability. Additionally, there are some approaches, differing from the idea of predefined reasoning structures, focus on how to decompose a complex problem into simpler problems for more effective resolution [14, 37].

Underlying these methods is the intuition of task decomposition and extensive empirical evidence has demonstrated its effectiveness. Furthermore, this inspires us to explore the potential of task decomposition in edge-cloud collaboration, allowing part of the task to be addressed by cloud-based models while the other part is handled by on-device models, thereby achieving a fine-grained collaborative framework.

### 2.2 On-device LLM Agent

The rise of LLMs has revolutionized AI applications, sparking widespread interest in personal AI assistants and mobile automation tools. As on-device intelligent agents gain popularity, users increasingly expect seamless, real-time AI support on their smartphones and devices. However, the limited computational and storage capabilities of edge devices pose significant challenges in deploying powerful models for such intelligent agents. Under these constraints, edge-cloud collaboration is a practical solution.

Apple's latest research [9] exemplifies this synergy by combining an efficient on-device model, AFM-on-device, with a powerful cloud-based model, AFM-server. This approach balances device limitations with the high-performance needs of Apple's AI features across their platforms. Similarly, [4] tackles edge-device limitations by splitting task planning and execution between two models—Octo-planner and Octopus—focused on efficiency and adaptability. Both approaches highlight the growing trend of edge-cloud collaboration to ensure powerful, low-latency AI experiences, though Apple leans more on cloud support while Octo-planner emphasizes on-device optimization.

Despite these advancements, a trade-off remains between the power of cloud models and the real-time performance requirements of on-device models. This transition towards edge-cloud collaboration stems from the recognition that edge devices, while increasingly capable, are still constrained in their ability to host and execute large-scale language models. The computational and storage limitations of such devices make it difficult to support models with 10 billion parameters or more, which often define state-of-the-art performance in many AI tasks.

## 3 Preliminaries

**Problem Definition** Denote the local deployed SLM as $\mathcal{M}_\mathcal{D}$, and the cloud-based LLM as $\mathcal{M}_C$. The user's original query is restricted to the edge model for task decomposition and allocation, while the resulting sub-tasks can be resolved by either $\mathcal{M}_\mathcal{D}$ or $\mathcal{M}_C$. For each

task $T$, we aim to devise an efficient task decomposition method:

$$T \rightarrow \{t^1, t^2, ... t^k\} \tag{1}$$

and an accurate model allocation strategy:

$$\mu : t^i \mapsto \{\mathcal{M}_\mathcal{D}, \mathcal{M}_\mathcal{C}\} \tag{2}$$

that prioritizes assigning simple subtasks to on-device model, while invoking the cloud-based model for handling more complex subtasks. Let the reasoning accuracy over the entire task set $\mathcal{T} = T_1, T_2, ... T_n$ be denoted as $Acc$, with the API cost represented by $C_{Api}$, and the completion time denoted as $C_{Time}$. Our objective is to minimize two types of costs while allowing for a controlled decrease in reasoning accuracy. Given an acceptable accuracy degradation threshold $\delta_{Acc}$, the optimization objective can be expressed as follows:

$$\min (C_{Api} + \lambda C_{Time}) \quad \text{s.t.} \quad Acc_{final} \geq Acc - \delta Acc \tag{3}$$

where $\lambda$ serves as a tunable weight.

## 4 Methodology

### 4.1 Division-of-Thoughts Framework

The *division-of-labour* is a prominent concept in economics, famously introduced by Adam Smith. It refers to the separation and allocation of tasks within any economic system or organization, enabling participants to specialize in specific areas based on their unique capabilities. This specialization allows individuals, organizations, and nations to optimize productivity by leveraging specialized skills, equipment, and resources. This concept inspires us to consider a similar approach in the context of edge-cloud collaboration, where user queries can be intricately divided: simpler sub-tasks can be assigned to SLMs while more complex ones are allocated to LLMs. This sub-task-level division of labor is expected to reduce reasoning costs while maintaining reasoning performance, enabling more efficient collaboration.

Building on this intuition, we developed the **DoT** edge-cloud collaboration framework. As illustrated in Figure 1, our framework is divided into three components. The first component is the *Task Decomposer*, which breaks down the user's query into several simpler and independent sub-tasks. The second component is the *Task Scheduler*, which is leveraged to determine the pairwise dependencies between sub-tasks and constructing a task dependency graph based on these dependencies. The third component is the *plug-and-play LLM adapter*, responsible for assigning each sub-task to the appropriate models. The adapter extracts sentence embeddings from the SLM and maps them to difficulty coefficients, which serve as the basis for the model allocation for sub-tasks. Importantly, the training of the adapter does not require modifying the LLM's parameters, ensuring that the LLM's question-answering remains unaffected. Once the appropriate models have been assigned to each sub-task, reasoning proceeds along the order defined by the constructed dependency graph, leading to the final results.

### 4.2 Decomposing user Query

As emphasized in the *division-of-labour*, an effective division method serves as the foundation for collaborative work. The granularity and accuracy of division directly influence the quality and efficiency of the collaboration. Previous work has explored the collaboration between large and small models, but the granularity of these efforts has often been limited to the query level, and most rely on manually designed task decomposition strategies. For instance, in the ToT approach to solving the 24-point game, the task is broken down into two sequential steps: proposing and evaluating, with proposing occurring before evaluating iteratively. However, this rigid and manual workflow is challenging to generalize across various tasks, which highlights the need for a flexible and fine-grained task decomposition method that operates without human intervention.

We have developed a fine-grained task decomposition method based on the powerful *In-Context Learning (ICL)* capabilities of LLMs. We leverage a meta-prompt that incorporates "chain-of-thought"-like prompting with hand-crafted task decomposing examples, to elicit the inherent *planning* abilities in LMs. For each benchmark, We randomly selected 8 samples, manually performed step-by-step task decomposition, and incorporated these steps into the prompts. To enable LLMs to independently solve each sub-task and effectively use the answers from preceding tasks as references for subsequent reasoning, we place great emphasis on the independence and clarity of the decomposed sub-tasks and have incorporated targeted cues in the prompts design.

### 4.3 Task scheduling via Dependency Graph

We employed a three-step approach to schedule all the sub-tasks effectively: **Dependency Judgement.** Prompting LLM agent to assess pairwise dependencies between sub-tasks. The prompts are as follows: *"Please list the dependencies in the format 'Subproblem A [xxx] -> Subproblem B [xxx]' indicating that Subproblem A must be completed before Subproblem B can start."* **Graph Construction.** Converting dependencies into a dependency graph. Based on the clear directional dependencies, constructing the graph is straightforward. However, we further trace back from the final sub-task node to calculate the depth of each sub-task node in the graph (analogous to the height in a tree structure). Sub-tasks at the same depth are independent of each other and can be inferred in parallel. Depth serves as the inference batch, with batches processed sequentially while tasks within a batch are reasoned in parallel. As shown in Figure 3, compared to sequentially reasoning through all sub-tasks, our graph structure can capture more precise inferential relationships while incurring fewer time costs. **On-Graph Reasoning**. We start by solving the sub-tasks from the shallowest depth batch, running the sub-tasks within the same batch in parallel. After completing one batch, we proceed to the next. During the reasoning of a specific sub-task, only the results of its prerequisite sub-tasks, rather than all previously solved tasks, are included in the prompts, which enables efficient graph-based reasoning.

### 4.4 Task Allocation with Plug-and-Play LLM Adapter

For the decomposed sub-tasks, we aim to assess their difficulty based on the task descriptions and allocate either cloud-based or edge-side models for execution. Using LLMs to evaluate difficulty introduces additional inference costs and often fails to accurately assess the sub-task's complexity, which motivates us to train a model specifically designed for task allocation with sentence embeddings.

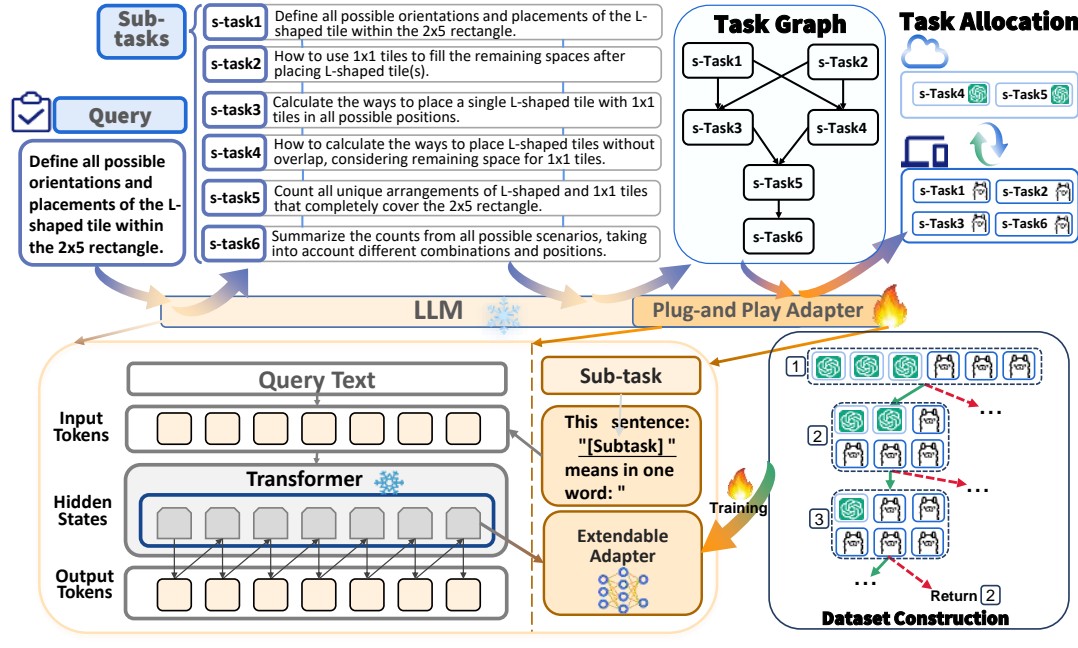

Figure 1: Whole framework of DoT.

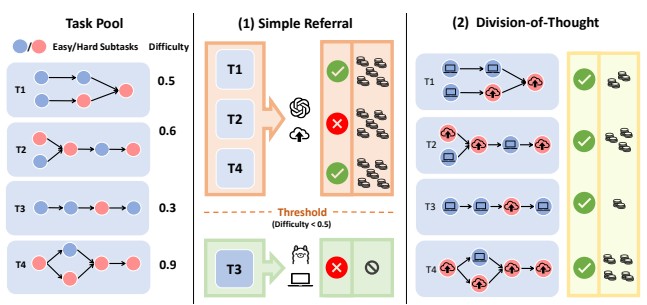

Figure 2: Advantages over the Simple Referral Paradigm

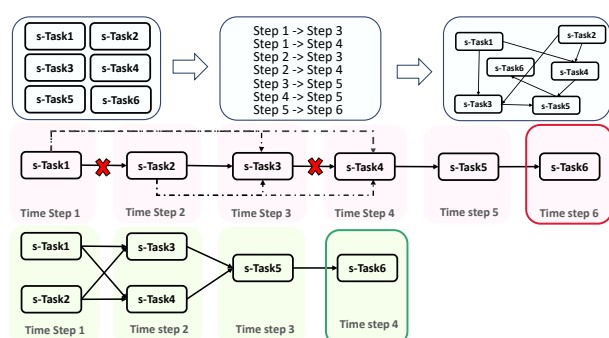

Figure 3: Details of Dependency Graph Construction [figure]

*Sentence embeddings*, which map a sentence to a fixed-size vector capturing its semantic meaning and context, have seen extensive application in natural language processing for its lightweight accessibility and the strong ability to capture sentence semantics. We can obtain a sentence embedding for each sub-task's prompt and then map the embedding to the corresponding difficulty. Given that LLMs are already deployed on the edge side and have been shown to serve as effective sentence embedders, we plan to leverage the local deployed SLM to produce sentence embeddings.

However, autoregressive LLMs lack specialized tokens such as BERT's [MASK] or [CLS], which are typically used in transformer-based models for embedding tasks. The exploration of Ting Jiang et al. [12] helps us overcome this limitation. Prompt-based method is refined specifically for autoregressive LLMs by instructing the model to generate the next word that captures the semantic meaning of the input sentence. Specifically, a simple but effective template can be constructed for each piece of text: *This sentence: "[text]" means in one word: "*, where *[text]* represents the input sentence,

and the LLM is prompted to generate the next token that encapsulates the core meaning of the sentence in a single word. The last generated token plays a critical role, as we extract its hidden state from the model and use it as the sentence embedding.

**Boosting SLM with Plug-and-Play Adapter.** Despite the varying lengths of sub-task descriptions, the sentence embeddings maintain a consistent dimensionality. For efficiency, we append a multi-layer perceptron (MLP) to the transformer module of the LLM to map the embeddings to a difficulty score. Given that both the edge side and the cloud host only one model, we can simply set the scores as 0 (simple, for the edge-side SLM) or 1 (complex, for the cloud-based LLM). The output from the MLP is translated into a model designation and passed to the agent handling the current reasoning step, providing a concrete allocation strategy for edge-cloud collaboration. Moreover, the adapter requires no modifications to SLM's parameters. It essentially serves as a flexible and extensible adapter for the SLM, avoiding the need to store two sets of parameters on the edge side.

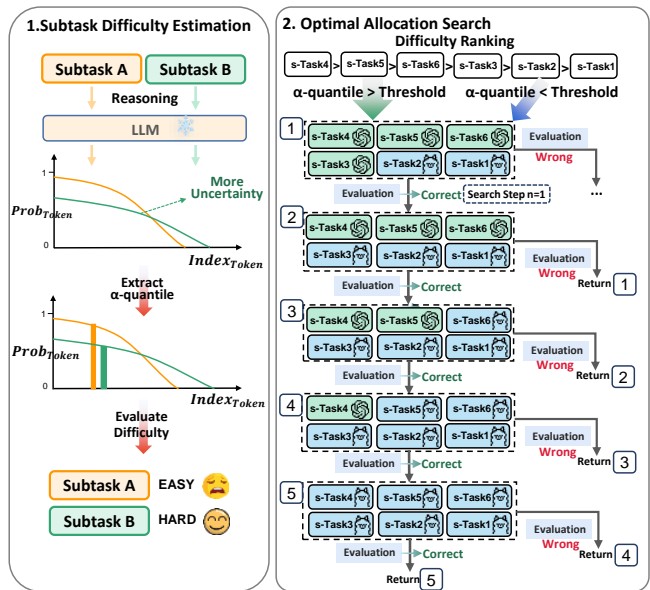

**Figure 4: Details of the Dataset Construction Method.**

**Creating Training Data with $\alpha$-Tree Algorithm.** Then how do we construct a dataset to train our adapter? We developed an efficient allocation optimization strategy, $\alpha - Tree$, which consists of two key components: the sub-task difficulty ranking based on the $\alpha$-quantile token probability, and a tree-based search strategy guided by the ranking, allowing us to generate a large-scale optimal allocation dataset with low cost and high speed.

$\alpha$-quantile refers to calculating a specific quantile from the token probabilities generated by LLM during inference. Unlike traditional methods that aggregate probabilities through summation or averaging, $\alpha$-quantile focuses on a specific portion of the uncertainty distribution—such as the minimum ($\alpha = 0$) or a higher percentile (e.g., $\alpha = 0.8$), which provides a fine-grained measure of uncertainty. Donate the input context as $x$, and the output tokens as $\hat{y}_i'$. The $\alpha$-quantile value can be denoted as:

$$s_{\text{quant}}(x, \alpha) \doteq \text{quantile}_\alpha \left( p^{(1)}(\hat{y}_1' \mid x), \ p(\hat{y}_2' \mid x, \hat{y}_1'), \ldots, \right.$$
$$\left. p(\hat{y}_n' \mid x, \hat{y}_1', \ldots, \hat{y}_{n-1}') \right)$$

In our implementation, we employed the SLM to answer all decomposed sub-questions, recording the sampling probability corresponding to each token. We then applied a consistent alpha value to extract the $\alpha$-quantile from each probability sequence and ranked the $\alpha$-quantile values for all sub-task responses. Since higher task difficulty leads to greater uncertainty in the model's answers, resulting in lower token sampling probabilities, we reversed the $\alpha$-quantile ranking to obtain the difficulty order of the sub-tasks.

Based on the $\alpha$-quantile values of the sub-tasks, we set a fixed allocation threshold: sub-tasks with values exceeding this threshold are assigned to SLM, while those below the threshold are assigned to LLM, forming the initial model allocation. From this initial allocation, we perform collaborative reasoning with SLM and LLM. If the reasoning result is correct, we reassign $n$ of the sub-tasks currently

handled by LLM—specifically those with the highest probability of being correctly answered—to SLM. Conversely, if the answer is incorrect, we transfer $n$ of the sub-tasks handled by SLM—specifically those with the lowest probability of being misanswered—to LLM. As shown in Figure 4, the structure of the search exhibits a tree-like form. The algorithmic workflow for dataset construction is illustrated in 1.

---

**Algorithm 1** Constructing Model Allocation Dataset

**Input:** Tasks which has been decomposed into several subtask:
$\left\{ T_1 = \{t_1^1, .., t_1^{k_1}\}, T_2 = \{t_2^1, .., t_2^{k_2}\}, ..., T_n = \{t_n^1, .., t_n^{k_n}\} \right\}$

**Output:** Fine-grained device-cloud model allocation scheme applied to each subtask within every task ($m_i^j \in \{\mathcal{M}_\mathcal{D}, \mathcal{M}_C\}$):
$\left\{ M_1 = \{m_1^1, .., m_1^{k_1}\}, M_2 = \{m_2^1, .., m_2^{k_2}\}, ..., M_n = \{m_n^1, .., m_n^{k_n}\} \right\}$

1: **for** task id i **do**
2:      Perform on-graph reasoning for eash subtask: $Answer = \{a_i^1, .., a_i^{k_i}\}$ with $Prob_{token} = \{p_i^1, .., p_i^{k_i}\}$
3:      Calculate the $\alpha$-quantile value for each token sequence: $V_{\alpha-quantile} = \{v_i^1, v_i^2, ..., v_i^{k_i}\}$
4:      Set a threshold $\theta$ to obtain the initial allocation strategy:
$$M_i = \{m_i^1, .., m_i^{k_1}\} \text{ where } m_i^j = \begin{cases} \mathcal{M}_\mathcal{D} & \text{if } v_i^j > \theta \\ \mathcal{M}_C & \text{if } v_i^j < \theta \end{cases}$$
5:      Reasoning on the task to obtain: *result = True or False*
6:      **while** True **do**
7:          **if** *result == True* **then**
8:              Find all subtasks assigned to $\mathcal{M}_C$, $M_{Ci} = \{m \in M_i \mid m == \mathcal{M}_C\}$, select N subtasks with the highest $\alpha$-quantile probabilities, and reassign them to $\mathcal{D}$. Get new $M_i'$.
9:          **else**
10:             Find all subtasks assigned to $\mathcal{M}_\mathcal{D}$, $M_{Di} = \{m \in M_i \mid m == \mathcal{M}_\mathcal{D}\}$, select N subtasks with the lowest $\alpha$-quantile probabilities, and reassign them to $\mathcal{M}_C$. Get new $M_i'$.
11:          **end if**
12:          Reasoning to obtain new result *result'*
13:          **if** $len(M_{Di}) == 0$ or $len(M_{Ci}) == 0$ or $result' \neq result$ **then**
14:             break
15:          **end if**
16:          $M_i = M_i'$
17:      **end while**
18: **end for**

---

## 5 Experiments

### 5.1 Experimental Setup

**Benchmarks.** We validated the effectiveness of our framework across seven benchmarks. These benchmarks are all publicly available and open-source. We categorized these benchmarks into four groups:

- **Logical Reasoning.** We classified the selected P3 and SCAN benchmarks into this category. These benchmarks place a stronger emphasis on logic, involving challenges

| Model | Logical Reasoning | | | | | | Web Browsing | | | Solving Math Problems | | | | | | | | | Commonsense Reasoning | | |
|---|---|---|---|---|---|---|---|---|---|---|---|---|---|---|---|---|---|---|---|---|---|
| | P3 | | | SCAN | | | WebShop | | | MATH | | | CHAMP | | | DROP | | | CSQA | | |
| | Acc | $C_{Time}$ | $C_{Api}$ | Acc | $C_{Time}$ | $C_{Api}$ | Acc | $C_{Time}$ | $C_{Api}$ | Acc | $C_{Time}$ | $C_{Api}$ | Acc | $C_{Time}$ | $C_{Api}$ | Acc | $C_{Time}$ | $C_{Api}$ | Acc | $C_{Time}$ | $C_{Api}$ |
| COT (GPT-4o) | **42%** | 35.8 | 4.45¢ | **68%** | 9.21 | 2.75¢ | 35% | 30.9 | 10.65¢ | 51.5% | 34.5 | 5.34¢ | 55.5% | 26.4 | 4.45¢ | 80% | 11.6 | 1.30¢ | 80% | 17.0 | 3.60¢ |
| TOT (GPT-4o) | 38% | 93.1 | 14.55¢ | 52% | 32.5 | 9.82¢ | **36%** | 62.4 | 47.34¢ | **63%** | 60.5 | 9.97¢ | 57% | 64.2 | 11.65¢ | 80.5% | 40.2 | 5.41¢ | 82% | 98.8 | 20.50¢ |
| COT (Llama 3-8B) | 5.5% | 18.1 | N/A | 17% | 5.0 | N/A | 0.0% | 10.5 | N/A | 10% | 21.1 | N/A | 19% | 13.1 | N/A | 72% | 3.8 | N/A | 70% | 8.4 | N/A |
| TOT (Llama 3-8B) | 5.5% | 58.3 | N/A | 13% | 21.8 | N/A | 1.4% | 22.5 | N/A | 29.5% | 49.0 | N/A | 25% | 68.1 | N/A | 65% | 27.8 | N/A | 68.5% | 89.4 | N/A |
| DataShunt | 14% | 25.1 | 2.45¢ | 23.5% | 7.6 | 1.72¢ | 34% | 30.9 | 8.35¢ | 16% | 24.9 | 1.66¢ | 34% | 19.1 | 2.98¢ | 74% | 8.6 | 0.60¢ | 73% | 10.4 | 1.28¢ |
| DoT (ours) | 41% | 23.5 | 1.58¢ | 63% | 5.5 | 1.20¢ | 31% | 17.2 | 4.97¢ | 59% | 22.6 | 1.02¢ | **58%** | 16.1 | 0.84¢ | **85%** | 4.9 | 0.32¢ | **82%** | 9.9 | 0.49¢ |
| Improvement | ↓2.38% | ↓34.36% | ↓64.49% | ↓7.35% | ↓40.28% | ↓56.36% | ↓13.89% | ↓72.43% | ↓89.95 | ↓6.35% | ↓62.72% | ↓89.50% | ↑1.75% | ↓74.92% | ↓92.79% | ↑5.59% | ↓87.81% | ↓94.09% | 0% | ↓89.98% | ↓97.60% |

**Table 1: Performance of DoT and baselines on 7 benchmarks. $C_{Time}$ and $C_{Api}$ are averaged expense for each task, where time consumption is measured in seconds, and api cost is measured in US dollar cents (¢). *N/A* appears in experiments where reasoning is conducted solely using LlaMA without invoking the OpenAI's API key. In each benchmark, the highest reasoning accuracy is highlighted in bold. The results of the baseline with the highest *Acc* are underlined which will be used to compute the "Improvement" in the last row.**

such as traversal, backward reasoning, and anomaly detection, which require a higher level of logical coherence and rigor.

- **Solving Math Problems.** This category of benchmarks primarily addresses mathematical problems, involving computation, mathematical knowledge, and problem-solving techniques. It is widely regarded as a crucial test of LLMs' reasoning abilities. We selected three benchmarks for this category: MATH, CHAMP, and DROP.
- **Commonsense Reasoning.** CSQA is a widely used commonsense reasoning dataset that places less emphasis on reasoning capabilities but requires a broader knowledge base. The difference in LLMs' parameter scales leads to disparities in the knowledge systems they possess, making CSQA well-suited for evaluating our collaborative reasoning scenario.
- **Web Browsing.** We used the WebShop benchmark for this category, which tests LLMs' reasoning abilities as interactive agents in real-world scenarios. It challenges agents to navigate web pages, interpret complex queries, and take appropriate actions to fulfill user requests. This benchmark assesses crucial skills for AI assistants, including understanding compositional instructions, reformulating queries, and strategically exploring information to meet specific requirements.

**Baselines and Prompting Methods.** Given the constraints of the edge-cloud collaborative setting, we established three baselines.

- **CoT.** This baseline decomposes the problem into sub-steps, solving each sequentially. We use a single LLM throughout, executing each step only once without iteration.
- **ToT.** ToT explores multiple solutions (M=5) for each sub-step. We use a scoring mechanism to retain the N=3 top-scored paths, with the highest-scored path determining the final answer. Like CoT, it employs a single LLM for the entire process.
- **DataShunt.** This approach dynamically selects between on-device small LLM and cloud-based large LLM at the start of each task. It prompts the LLM to evaluate task complexity, assigning a *solving model* that is then used consistently for all sub-steps.

In the task decomposition processes involved in CoT and ToT, we employed the same prompting method used in our DoT, applying eight hand-crafted examples to each benchmark. These examples were randomly sampled from the dataset and are orthogonal to the test set. The specific prompts can be found in Appendix A.5.

**Selection and Deployment of LLMs** We used GPT-4o [26] as cloud-based LLM and Llama 3-8B [24] as on-device LLM as its parameter scale is theoretically deployable on edge devices. The Llama 3-8B model is deployed on a local A100 GPU with a context length limit of 8192. The parameter count of our adapter is only 13,109,249, approximately 1/800th of the parameters in local Llama.

**Evaluation.** For evaluation, we established three metrics: *Acc*, $C_{time}$, $C_{Api}$. Six of the benchmarks, excluding WebShop, have deterministic results. For instance, the results for CSQA are single-letter options, allowing direct determination of correctness. Similarly, for P3, the reasoning output can be passed into a problem function, and if the function returns True, the reasoning is deemed correct. It is worth noting that, although benchmarks like MATH and CHAMP have deterministic results, the presence of mathematical formulas complicates direct evaluation. Therefore, we rely on a large language model (GPT-4o) to verify the results, but the time and token costs for evaluation are not included in the total cost. For WebShop, we use the evaluation environment provided by the original benchmark to average the reward of all purchased items on the level of satisfaction according to the constraints.

## 5.2 Main Results

Comparison between our method and the baselines are shown in Table 1, we have highlighted in bold the highest accuracy results among the five baseline experiments on each benchmark, while the associated costs are underlined. We computed the relative change in our results compared to the baseline with the highest accuracy.

The experimental results demonstrate that our approach significantly reduces costs while maintaining accuracy within an acceptable range of decline. Across seven benchmarks, the relative change in accuracy compared to the best baseline results was: -2.38%, -7.35%, -13.89%, -6.35%, 1.75%, 5.59%, and 0%. Notably, positive values were observed in some cases, indicating potential improvements in reasoning accuracy, which suggests that our framework enhances the reasoning capability of LLMs. This enhancement will be further

| Methods | SLM Ratio | SR | # Evaluation | API Cost |
|---|---|---|---|---|
| Zero-Shot LLM | 53.11% | 92.78% | 1 | $2.56 |
| Binary-Search | 69.46% | 88.45% | 8.456 | $16.45 |
| $\alpha$-tree (n=1) | 85.53% | 99.44% | 3.4589 | $7.34 |
| $\alpha$-tree (n=2) | 86.45% | 96.34% | 2.3445 | $5.12 |

**Table 2: Methods for Searching the Optimal Allocation Scheme**

discussed in ablation study 5.4. At the same time, our approach achieved a substantial reduction in cost compared to the baseline with the highest accuracy, with an average time reduction of 66.12% and an average Api cost reduction of 83.57% which far exceeded the decline in accuracy.

Moreover, on P3 and SCAN benchmarks, CoT achieved the best performance, indicating that a sequential linear reasoning structure is better suited for these task types. In contrast, on more complex mathematical reasoning benchmarks like MATH, CHAMP, and WebShop, ToT achieved the highest reasoning accuracy, demonstrating the effectiveness of the multiple-proposal strategy. However, our approach incurred less than half of CoT's cost and only one-tenth of ToT's cost while still achieving comparable performance, highlighting the broad applicability of our graph-structured reasoning and the effectiveness of the edge-cloud collaborative framework.

To evaluate whether our collaborative approach fully leverages on-device models for reasoning, we calculated the proportion of reasoning time by the small on-device model relative to the overall task execution time, as well as the percentage of sub-tasks assigned to the edge-side model. These metrics were compared against the baseline *DataShunt*, and the results are presented in Figure 5. As shown in the figure, across all benchmarks, our allocation strategy utilizes on-device models more effectively, with the reasoning time and task allocation percentages exceeding those of the baseline by 11.99% and 21.92%, respectively.

## 5.3 Efficiency and Quality of Dataset Construction.

To validate the effectiveness of our method for difficulty assessment and optimal allocation search in dataset construction, we compared our $\alpha$-tree with other baseline search methods. We established two baseline methods: **Zero-shot LLM**: This approach utilizes a LLM to assess the difficulty of sub-tasks. The evaluation model employs GPT-4o, utilizing a question-and-answer format that incorporates the task content, all sub-tasks, and the current sub-task into the prompt. **Binary-Search**: This method employs a binary search strategy. Initially, all sub-tasks are processed uniformly using GPT-4o. If the reasoning results are correct, half of the sub-tasks are randomly assigned to Llama 3-8B, with this random assignment repeated N times until successful. In the second round of binary search, half of the remaining sub-tasks assigned to the larger model are again randomly allocated to the smaller model. This process continues until all sub-tasks are either handled by the smaller model or incorrect reasoning persist. We set N to 5.

For our $\alpha$-tree approach, which sequentially searches for optimal solutions according to difficulty, we tested with n=1 and n=2 (where

n represents the number of models whose allocation is altered at each search step). The comparison metrics included two quality assessment indicators: *SLM Ratio*: the proportion of the smaller on-device model in the final allocation scheme, and *SR*: success rate of reasoning with the final allocation scheme applied. Additionally, we considered two cost metrics: *# Evaluation*: the number of search iterations, where a single search requires reasoning for all sub-tasks of the entire task, and *Api Cost*, which refers to the expense incurred when invoking the cloud-based large LLM API during dataset construction.

We visualize the quality and cost of the constructed dataset on the MATH benchmark. As shown in Table 2, in terms of construction quality, our $\alpha$-tree achieved the highest proportion of small model usage while maintaining the highest reasoning accuracy. This demonstrates that our method finds a more efficient collaborative allocation strategy, ensuring high-quality reasoning while minimizing inference costs. *Zero-shot LLM* resulted in the lowest proportion of SLM usage, indicating that LLMs fail to assess the difficulty of sub-tasks. *Binary-Search* also showed a lower SLM proportion compared to ours. While increasing the number of search attempts (N) could potentially yield better results through random search, this unordered and purely random strategy will incur significant costs. Even with the current 5-time search, the cost is already several times higher than our approach. Further scaling will bring unsustainable cost. Naturally, *LLM-Eval* has the lowest cost, but its construction quality is significantly inferior.

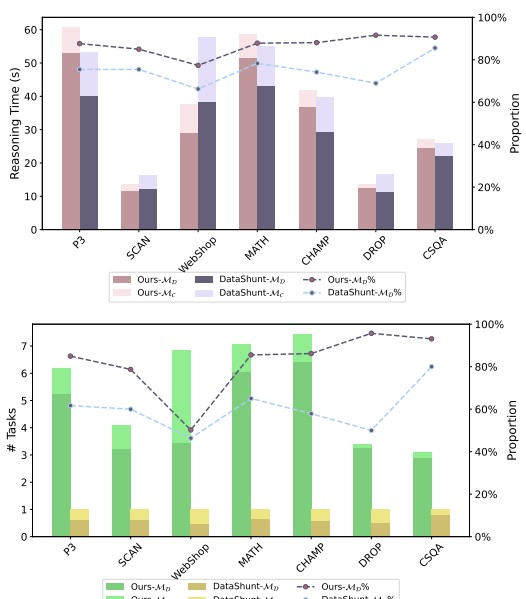

**Figure 5: Proportion of small models in reasoning time and tasks.**

## 5.4 Ablation Study

To validate the effectiveness of each design aspect of our model, we conducted ablation studies focusing on two key components. The first ablation removes the dependency graph construction and

| Model | Logical Reasoning | | | | | | Web Browsing | | | Solving Math Problems | | | | | | | | | Commonsense Reasoning | | |
|---|---|---|---|---|---|---|---|---|---|---|---|---|---|---|---|---|---|---|---|---|---|
| | P3 | | | SCAN | | | WebShop | | | MATH | | | CHAMP | | | DROP | | | CSQA | | |
| | Acc | Time | Api | Acc | Time | Api | Acc | Time | Api | Acc | Time | Api | Acc | Time | Api | Acc | Time | Api | Acc | Time | Api |
| DoT w/o Graph | 38% | 36.4 | 1.44¢ | 57% | 5.2 | 1.18¢ | 31% | 20.7 | 4.97¢ | 56% | 20.9 | 0.99¢ | 52% | 27.7 | 0.82¢ | 80% | 4.2 | 0.26¢ | 77% | 9.6 | 0.48¢ |
| DoT w/o Allocation | 43% | 39.8 | 4.80¢ | 64% | 10.7 | 3.66¢ | 39.6% | 19.4 | 13.89¢ | 78% | 36.9 | 7.68¢ | 64% | 30.4 | 5.05¢ | 84% | 12.8 | 1.69¢ | 85% | 20.6 | 1.49¢ |
| DoT (ours) | 41% | 23.5 | 1.58¢ | 63% | 5.5 | 1.20¢ | 31% | 17.2 | 4.97¢ | 59% | 22.6 | 1.02¢ | 58% | 16.1 | 0.84¢ | 85% | 4.9 | 0.32¢ | 82% | 9.9 | 0.49¢ |

**Table 3: Result of ablation study.**

on-graph reasoning, replacing it with a sequential reasoning resolution. The second ablation removes the model allocation mechanism, instead using the cloud-based LLM exclusively for reasoning. Notably, the second ablation experiment provides a direct assessment of whether our reasoning workflow can enhance the reasoning performance of LLMs.

The experimental results are presented in Table 3. We were pleasantly surprised to discover that, without considering reasoning costs, using the large cloud-based model exclusively for reasoning can greatly improve the reasoning accuracy. This improvement was particularly significant on the challenging MATH and CHAMP benchmarks, where accuracy increased from 63% to 78% and from 57% to 64%, respectively, compared to the *ToT (GPT-4o)*, which had the highest accuracy among baselines. Improvements on other benchmarks were relatively limited. Understandably, relying solely on the cloud-based model results in several times the computational cost compared to the collaborative approach, but this cost remains acceptable when compared to ToT. Even when using the smaller model exclusively, its accuracy was significantly improved over *CoT (Llama 3-8B)* and *ToT (Llama 3-8B)*. For example, accuracy on Puzzle increased from 5.5% to 29%, and on SCAN from 17% to 52%. Additionally, because the smaller model is deployed on edge devices, the overall cost was greatly reduced. Additionally, the ablation results after removing the dependency graph construction and on-graph reasoning (*DoT w/o Graph*) further demonstrate the critical role of the graph structure in reasoning.

### 5.5 Trade-off Between Accuracy and Cost.

The ablation study demonstrates that our on-graph reasoning method significantly enhances the reasoning capability of LLMs, though at a higher computational cost. Offloading all sub-tasks to the powerful cloud-based models can achieve higher reasoning accuracy. In this section, we delve into the detailed relationship between cost and reasoning accuracy to identify the achievable performance upper bound under different budget constraints. Based on our original task allocation strategy, we generate diverse model allocation schemes by proportionally assigning more sub-tasks to either the cloud-based model or the on-device model according to the difficulty level of each sub-task, resulting in the various data points shown in the figure. We also introduce *DataShunt* as a baseline, where the proportion of tasks handled by the cloud-based models is adjusted by tuning the LLM evaluation thresholds. We conducted experiments on four benchmarks, plotting the trade-off curves between accuracy and API cost. As illustrated in Figure 6, our curve consistently remains above the baseline curve. Under the same cost,

our approach demonstrates higher reasoning accuracy. We also achieves a higher upper bound on reasoning accuracy.

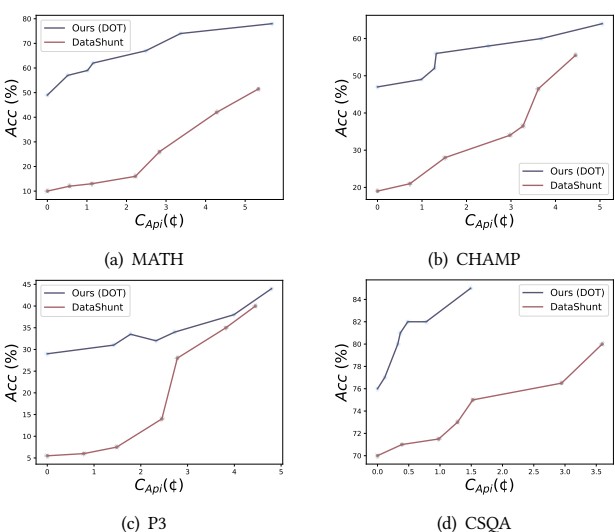

(a) MATH  (b) CHAMP

(c) P3  (d) CSQA

**Figure 6: Acc − $C_{Api}$ trade-off curves on 4 benchmarks.**

### 6 Conclusion

In conclusion, our proposed *Division-of-Thoughts* collaborative reasoning framework effectively enhances the reasoning performance of LLMs on edge devices by intelligently distributing tasks between local and cloud resources. The framework not only breaks down complex tasks into manageable subtasks but also leverages a directed-acyclic graph for optimal scheduling, maximizing reasoning accuracy while minimizing input context. Furthermore, the novel task allocator significantly improves local model performance with minimal memory overhead, making it well-suited for resource-constrained environments. Looking ahead, we aim to incorporate additional cloud-based large models, tailored for specific tasks, to refine the overall collaborative framework. We also intend to integrate a broader array of updated benchmarks to thoroughly evaluate the framework's performance across diverse scenarios, ultimately advancing the state of edge AI applications.

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

# Appendix

# A Introdcution of Benchmarks

## A.1 Mathematics

- **MATH** [11]. Mathematics Aptitude Test of Heuristics (MATH), comprises 12,500 problems from prestigious U.S. mathematics competitions like AMC and AIME. These problems, collected from platforms such as AoPS, test advanced problem-solving skills beyond standard K-12 math. Each problem includes a step-by-step solution and a final answer. The dataset spans seven subjects: Prealgebra, Algebra, Number Theory, Counting and Probability, Geometry, Intermediate Algebra, and Precalculus, with difficulty levels ranging from 1 (easy) to 5 (challenging), allowing models to learn and apply various mathematical heuristics.

- **CHAMP** [22]. Concept and Hint-Annotated Math Problems (CHAMP). This benchmark features 270 non-routine, competition-level problems sourced from Engel's Problem-Solving Strategies. The problems span five categories: number theory, polynomial, sequence, inequality, and combinatorics, requiring creative strategies and specific tricks. Each problem includes a final checkable answer and a step-by-step solution in natural language. The dataset contains 54 concepts and 330 hints, averaging 1.4 concepts, 1.7 hints, and 6 solution steps per problem. Problem statements average 20.2 words, with solutions averaging 10.9 words per step, highlighting the dataset's complexity and challenge.

- **DROP** [7]. Discrete Reasoning Over Paragraphs (DROP), is an English reading comprehension dataset with 96k adversarially-created questions. It challenges systems to resolve references in a question and perform discrete reasoning operations like addition, counting, or sorting over multiple input positions within a paragraph. The dataset is crowdsourced and derived from Wikipedia passages designed for complex questions. DROP demands a deeper understanding of paragraph content than previous datasets, with rigorous validation to ensure the quality of its development and test sets.

## A.2 Logic

- **P3** [27]. Python Programming Puzzles (P3), introduces a new type of programming challenge for evaluating program synthesis. P3 contains 397 Python-based puzzles, where the goal is to find an input that makes a given function return True. The puzzles span various difficulty levels, from simple string manipulations to complex algorithmic problems. P3 solvers, such as GPT-3 and Codex, can solve puzzles without reference solutions by learning from past attempts, with Codex achieving the highest success rate. P3 is designed for objective and comprehensive evaluation across a wide range of programming tasks.

- **Scan** [17]. This benchmark is used to evaluate the sequence-to-sequence learning ability to translate simplified natural language commands into action sequences. SCAN contains 20,910 commands generated by a phrase-structure grammar, which describe basic actions such as "jump" or "walk"

and their combinatorial variations (e.g., "jump around left"). While the dataset was originally designed to test a model's ability to generalize to unseen commands by applying learned interpretation functions, the logical structure and constraints involved in the translation process also make it ideal for assessing LLM reasoning capabilities.

## A.3 Commonsense

- **COMMONSENSEQA** [30]. This benchmark is a challenging dataset designed to test commonsense question answering. It consists of 12,247 multiple-choice questions created using concepts from CONCEPTNET. Each question is authored by crowd-workers to differentiate between multiple target concepts that share a semantic relation with a source concept, encouraging the use of prior knowledge and complex reasoning. COMMONSENSEQA pushes beyond simple associations, making it a more difficult task compared to traditional question-answering benchmarks.

## A.4 On-device AI assistant application

- **Webshop** [35]. WebShop serves as a simulated e-commerce environment featuring 1.18 million real-world products and 12,087 crowd-sourced text instructions. Designed to evaluate language agents, it challenges them to navigate diverse web pages and perform actions based on natural language product specifications. Agents encounter obstacles such as interpreting compositional instructions, reformulating queries, and understanding noisy text on webpages, while strategically exploring to fulfill product requirements. The modular design of WebShop separates website navigation from task-specific elements, allowing for easy adaptation to new tasks and domains. This dataset provides a robust platform for assessing the capabilities of language agents in an interactive, real-world-inspired setting, emphasizing their ability to comprehend and act on complex instructions.

## A.5 Implementation Details across Benchmarks

*A.5.1 MATH.* For MATH tasks, we employ a structured workflow comprising four key stages: Task Decomposition, Model Allocation, Dependency Graph Construction, and Step-by-Step Reasoning Based on the Graph. In the task decomposition phase, we prompt the LLM with exemplars of manually decomposed complex problems. The LLM is then instructed to generate manageable subtasks that collectively solve the primary challenge. Subsequently, we task the LLM with establishing subtask dependencies, where a relationship $Step_i \rightarrow Step_j$ indicates that $Step_i$ must precede $Step_j$. Using the derived dependencies, we construct a reasoning graph via Breadth-First Search (BFS). Subtasks at the same depth are processed in parallel. Upon completion of all subtasks, we conduct a final query to obtain the ultimate solution. This solution undergoes LLM-based evaluation through comparison with the ground truth.

*A.5.2 CHAMP.* For CHAMP tasks, the process is largely similar to the MATH process with the primary distinction being the specific few-shot examples of human-written decompositions provided to the LLM.

*A.5.3  DROP.* For DROP tasks, we adapt our approach to accommodate the format of questions based on given texts. The prompt for each step incorporates relevant background information provided in the dataset while maintaining the core implementation structure used in other benchmarks.

*A.5.4  P3.* The Programming Puzzle tasks present a unique challenge, requiring the LLM to generate inputs that yield a 'True' output for a given function. In addition to the puzzle description, we provide the LLM with the expected data type of the final input. The evaluation process for P3 differs from other benchmarks: we execute the program using the LLM-generated input and assess the correctness based on the program's output.

*A.5.5  SCAN.* For SCAN tasks focused on translating natural language instructions into action sequences, the steps until evaluation stay the same. The final phase involves converting the model's natural language outputs into standardized action sequences using few-shot examples. The evaluation is conducted by directly comparing these sequences with the true answers, ensuring the outputs accurately match the expected actions.

*A.5.6  COMMONSENSEQA.* For CSQA tasks, presented as multiple-choice questions based on common sense, our implementation has an identical structure as other tasks, while both the problem and its options are presented to the LLM for the following task decomposition and reasoning. The Final Evaluation consists of the LLM choosing the most plausible answer from the provided options. This selected answer is cleaned and will directly compare with the correct choice.

*A.5.7  Webshop.* The WebShop task presents unique challenges due to its interactive nature, where each action influences subsequent states and available options. Unlike our other experiments, WebShop is not perfect for the construction of a complete, predefined reasoning graph. Instead, our framework dynamically generates and executes sub-tasks based on the current state of the shopping session, in which the shopping process is divided into high-level sub-tasks/thoughts. We utilized the Model Allocation for each step when generating an action, either be 'think[]', 'click[]', or 'plan[]'. The first step would be to generate a keyword from the instructions to search on the website. Then, the top $N = 10$ matched items are recorded for the following procedures. With the conversation from the webpage, we implemented the task decomposition process to get a roadmap for the following actions. Prompting each step at once to the LLMs, the detailed information about each item will be recorded in the search history. Then, an evaluation and comparison process is prompted to choose the best item from the list. Lastly, the LLM will be prompted dynamically based on the previous steps, and self-navigated to complete the final purchase. This implementation demonstrates our framework's flexibility in handling tasks with dynamic, state-dependent decision-making processes.

# B  Prompts
## B.1  Task Decomposition

I will now give you a [Based on the type of problem]. The type of problem is type. Please break this problem down into several easy-to-solve steps.

1 examples are as follows: [Manual Written Examples]

Now the command is question, please decompose it into easy-to-solve steps like the examples. Answer Format: (Please write each broken-down question step on a separate line, starting with a number.)

To solve the question "xxx", we need to know: "1. question $step_1$", "2. question $step_2$", "3. question $step_3$". ...

## B.2  Dependency Construction

*System Prompt:*
Now we have a problem, which we have broken down into many sub-problems. I want you to understand the connection between these sub-problems

*User Prompt:*
The init problem is question. And the sub-problems are steps. Please provide your understanding of the relationships between these sub-problems. Your response must be concise.

Now we need to create standardized connections for the relationships between these sub-problems. Now Given the following subtasks for question: question, determine the dependencies between them:

[List of Steps]

Please list the dependencies in the format 'Subproblem A [xxx] -> Subproblem B [xxx]' indicating that Sub-problem A must be completed before Sub-problem B can start. Please identify any potential conditional dependencies from a logical perspective.

Answer format: (Please strictly follow the format. Each dependency should be separated by a new line. No explanation is required.)

Step $ID_i$ [ sub-problem i ] -> Step $ID_j$ [ sub-problem j ]
Step $ID_j$ [ sub-problem m ] -> Step $ID_n$ [ sub-problem n ]
...

## B.3  Subtask Reasoning

*System Prompt:*
Here is a math word problem. I will first provide a passage of the problem to set the context. Then, I will ask a specific question that requires you to use the information from the problem description, along with calculation and reasoning, to solve it.
Passage:
{ passage }
Question:
{question}

I have broken this math question down into several smaller questions. I will assign you sub-questions one by one, and provide the results of the previous sub-questions as a reference for your reasoning. Please solve the question according to mathematical logic.

*For each steps* So far, the answers to the resolved sub-questions are as follows: The format is Sub-question-Id: xxx; Sub-question:

xxx; Answer: xxx. Sub-question-Id: [Corresponding ID]; Sub-question: [Corresponding Step]; Answer: [Corresponding Solution for the step]

Among them, sub-questions predecessors are directly related to this sub-question, so please pay special attention to them. The sub-question to solve now is xxx: {subtask} Based on the information above, please provide a concise and clear answer

### B.4 Final Answer Conclusion

Now that all the sub-questions have been solved, so what is the final answer? Please give the final answer without any additional explanation or clarification.

### B.5 Model-Eval

Here is a [Problem type] problem with a standard answer and a student's solution. Please help me determine if the student's solution is correct.
Problem: {question}
    Standard answer: {True Answer}
    Answer: {Answer concluded by Model}

    If the student's answer is correct, just output True; otherwise, just output False. No explanation is required.

### B.6 ToT Evaluation

Please provide a confidence rating for the accuracy of this solution, on a scale from 1 to 5. Only output the number.

### B.7 SCAN Action Sequence Mapping

Now I have a pseudo action sequence expression with parentheses and multiplication. I need you to help me convert this into a sequence of actions without an operator sign.
    [Examples of mapping procedure]
    The pseudo action sequence to be converted is as follows: sentence Please change it to the action sequences.
    Please JUST answer the result.

### B.8 Webshop Prompts

Given the special characteristic of Webshop tasks as requiring a dynamic interaction with the shopping environment, we made a unique prompting structure for this task.

#### B.8.1 System Role

. You are an online webshop agent. You are instructed to complete a shopping process. You have to generate the next step in the process. Your action should take into account the most current observation (which is the page you are on) and the previous actions taken. Note: If no item absolutely meets the requirement, choose the one that meets most requirements.
    There are three types of actions you can output:
    1. search[query]: You can search for a product based on the query. The query is a string that describes the product you are looking for.
    2. think[thoughts]: You can think about the current state of the shopping process and decide what to do next.
    3. click[button]: You can click on a button on the page to navigate to another page. Where the button are presented in the observation

that is bracketed by []. If you think a product is the best choice, you can click on the "Buy Now" button to end the process.
    Example of a valid output:
search[noise cancelling cosy cost usb microphone]
think[I want to compare the features of the products]
click[Buy Now]
click[< Prev]
Note Don't output Action in front of the action. The action should be in the format of [action][content].

#### B.8.2 Webshop Task Decomposition

. I have an online shopping request with some constraints and I need to find the best options, and I have searched for the key words with some top results. You should help me to decompose the question into sub-steps that should be done for the following process. You will have the tools to help you with the online shopping.

    Here is a example of decomposed tasks: Given the information of the current state 1 Action: reset Observation: WebShop Instruction: i want a noise cancelling cosycost usb microphone, and price lower than 60.00 dollars [Search]

    Action: search[noise cancelling cosycost usb microphone] Observation: [Back to Search] Page 1 (Total results: 50) [Next >] [B0972Q1T8T] Cosycost USB Microphone, Condenser Computer PC Gaming Microphone for PS4/5 Laptop Windows Mac OS Android Phone,Noise Cancelling Instant Mute,Studio Mic for Voice,Music Recording, Podcasting,Streaming $32.99 [B072L2D6LY] Andrea Communications NC-255VM USB On-Ear Stereo USB Computer Headset with Noise-Canceling Microphone, in-Line Volume/Mute Controls, and Plug $34.59 [B071H84LTJ] Andrea Communications NC-455VM USB Over-Ear Circumaural Stereo USB Computer Headset with Noise-Canceling Microphone, in-Line Volume/Mute Controls, and Plug $49.24

    Example of decomposed tasks:
    To solve the question, we need to clarify/solve:
1. Click and check item B0972Q1T8T to get more detailed information.
2. Click and check item B072L2D6LY to get more detailed information.
3. Click and check item B071H84LTJ for more information.
4. Based on the more detailed information, I will compare to see which one fulfill by request.

    Now, the current state is {prompt}
    Based on the current process, please decompose it into sub-steps. Make sure to answer in this format Answer Format: (Please write each broken-down question on a separate line, starting with a number.) To solve the question "xxx", we need to clarify/solve:
"1. Click and check xxxx",
"2. Click and check xxxx",
"3. sub-question 3".

#### B.8.3 ToT Evaluation

. Please evaluate the following shopping process based on these criteria: 1. Relevance to the original request 2. Efficiency of the search and decision-making 3. Comparison of multiple options 4. Attention to product details and customer requirements
    Background information: {init prompt} {prompt}
    Shopping process: {search action}

Rate the overall quality of this shopping process on a scale from 1 to 10, where 10 is the best. Only provide the numerical score as your answer.

### B.8.4  Reasoning

. Here is an example of the shopping process:

[One Manual Written Example]

This is the current instruction:

Instruction: [Shopping Request]

Action: XXX

Observation: XXX

(Here The Action and Observation will be updated dynamically based on the action generated by LLMs)

