# OpenReview forum: "Division-of-Thoughts: Harnessing Hybrid Language Model Synergy for Efficient On-Device Agents"
_ACM.org/TheWebConf/2025/Conference — WWW 2025 Oral_

### Official Review · Reviewer_Am9u · 2024-11-22

**Novelty:** 5
**Technical Quality:** 5

**Review:**

The authors propose a method to distribute language model tasks to small and large language models to minimize cost and latency of the user being served. They use LLMs to break down the task and propose a new task scheduling approach via dependency graph. They allocate tasks via a "plug-and-play" adapter. They show that they can save costs with their method, but with the downside of some accuracy gain (5.6%) but also significant accuracy drop (up to almost 14% depending on the task).

**Pros:**

Scheduling tasks between small and large LMs, and their trade-offs between accuracy and costs, is an important task application.

Their thresholding based approach for task distribution seems original .

The paper is well written.

Thorough ablation studies.

Evaluation on many tasks (math, commonsense, browsing, reasoning).


**Cons:**

Relying on GPT-4o for evaluation seems entangled if it is also used as the cloud based LLM (Section 5.1). How can the same model be used as the provider of the answer and the evaluator?

Capturing the core meaning of a sentence in one word, and then using its hidden state seems potentially too simple to capture an actual difficulty score. Was there any evaluation on why such a simplification is necessary? Also the authors should mention this as a limitation, as their adapter may need to be re-trained based on different architectures producing the hidden state.

Line 695, the claim that this approach enhances the reasoning capabilities of LLMs is a big claim, especially when only evaluated on two models. Is there more data to back it up?


**Other improvements:**

Captions of figures are not self-contained. It is often hard to find what each plot is supposed to show by combing through the text. Sometimes, figures are not even mentioned in the text (e.g. Figure 2)

Figure 3, the claim of more precise inferential relationships is not backed by quantitative data, or at least not mentioned in the text.

Algorithm 1 can be cleaned up to be more readable, e.g. replacing while true utilizing the if ... break  condition (line 559).

Please check the use of acronyms and that they are introduced before they are used (e.g. AFM, SLM, ...).

Datasets and baseline prompting methods that were used are not properly cited (Section 5.1).

Information should be given on which LLM was used for zero shot in Table 2 .

Typo in line 502, Donate should be Denote

**Questions:**

**Other questions:**

Related work should mention how their approach addresses some downsides mentioned (e.g. line 182/183). How does their approach improve generalizability?

Is your optimization objective (Eq. 3) connected to the results shown? E.g. is there a study focusing on different accuracy drops and how much can be gained with your approach?

**Reviewer Confidence:**

3: The reviewer is confident but not certain that the evaluation is correct

**Scope:**

4: The work is relevant to the Web and to the track, and is of broad interest to the community

---

### Official Review · Reviewer_J64y · 2024-11-28

**Novelty:** 4
**Technical Quality:** 4

**Review:**

This paper proposes a new collaborative reasoning framework, Division-of-Thoughts (DoT), that leverages locally deployed small language models (SLMs) and large language models (LLMs) in the cloud to enable efficient large language model agents on edge devices through task decomposition and optimized subtask allocation.

#### Pros

1. An efficient task decomposition and collaboration framework for edge computing is proposed, which is innovative.
2. Excellent results in cost-performance balance are achieved, which is suitable for large-scale commercialization.
3. The experiments cover multiple benchmark datasets with detailed result data.

#### Cons

1. The method relies heavily on existing technologies and needs to be further strengthened for originality.
2. The computational overhead and scalability of key modules (e.g., dependency graphs, adapters) are under-described.
3. The applicability of the framework in different hardware environments and dynamic task scenarios is not tested.

**Questions:**

1. How are dependencies between subtasks identified? If there is a dependency identification error, does it significantly affect the final performance?
2. Has the impact of the number and depth of task nodes in the dependency graph on inference efficiency been evaluated?
3. Does the adapter depend on a specific model architecture (e.g., Llama 3-8B)? Can it be adapted to architectures other than Transformer?
4. What is the performance of the adapter if hardware resources are more limited (e.g., low-end cell phones)?
5. How is the value of α chosen in the α-tree algorithm? Does a generalized tuning strategy exist?

**Reviewer Confidence:**

3: The reviewer is confident but not certain that the evaluation is correct

**Scope:**

3: The work is somewhat relevant to the Web and to the track, and is of narrow interest to a sub-community

---

### Official Review · Reviewer_jcWL · 2024-11-30

**Novelty:** 5
**Technical Quality:** 5

**Review:**

This paper proposes a cloud-edge collaboration strategy Division-of-Thoughts (DoT) for LLM agents-driven knowledge reasoning. In DoT, a meta-prompt based decomposer first decomposes the task into several sub-tasks, which will be allocated either to SLM in the edge device or LLM in the cloud server. An adapter is trained to assign the sub-tasks. The experiments show DoT achieves a better trade-off between accuracy and efficiency. The research topic is quite relevant to the WebConf.

**Questions:**

Strength:

S1. A novel cloud-edge collaboration framework is proposed.

S2. The experiments are conducted on four types of LLM reasoning tasks.  It's appreciated that the authors made a large effort in the experiment to test the cloud-edge collaboration framework of GPT4o and Llama.

S3. The presentation is generally clear. The paper is well organized with clear motivations.



Weakness:

W1. The idea of dependency graph and relative formulations is straightforward.

W2. The optimization objective in the performance is not fully consistent with that in the algorithm and the evaluation metric.

W3. The paper should add more details on the alpha-quantile regression, which may be a key technical contribution in this paper.


Details:

D1. The optimization objective is a linear combination of task complete time and the API cost. The algorithm does not directly optimize this objective so it is unclear how this objective function changes in the overall training process.

D2. In the experiments, the authors compare the API cost and the task complete time as two separate metrics, which is also inconsistent with the objective optimization. How do the changes of \lambda influence the performance of DoT? The author should conduct a parameter analysis on that.

D3. DL models generally suffer from the problem of uncertainty quantification. Whether and how the alpha quantile regression here can help to relieve this problem?

**Reviewer Confidence:**

3: The reviewer is confident but not certain that the evaluation is correct

**Scope:**

4: The work is relevant to the Web and to the track, and is of broad interest to the community

---

### Official Review · Reviewer_h31F · 2024-12-01

**Novelty:** 5
**Technical Quality:** 4

**Review:**

This paper presents Division-of-Thoughts (DoT), a novel framework for efficient edge-cloud collaboration in LLM-based AI assistants. The framework intelligently decomposes complex queries into subtasks and allocates them between locally deployed smaller language models (SLMs) and cloud-based large language models (LLMs) based on task difficulty. The key innovation is the use of a graph-based task scheduler and a lightweight plug-and-play adapter that helps determine optimal task allocation without modifying the base model parameters.

The pros and cons of the papers are:
PROS:
1. Technically good framework design: The paper presents a well-structured solution to edge-cloud collaboration through a novel "division of labor" approach. The three-component architecture (Task Decomposer, Task Scheduler, and Plug-and-Play Adapter) provides a clear separation of concerns and addresses key challenges in deploying LLM agents on edge devices. The self-reinforced training method for the adapter is innovative as it eliminates the need for human annotation while maintaining model performance.

2. Strong empirical performance: The framework achieves substantial practical improvements, demonstrating average reductions of 66.12% in reasoning time and 83.57% in API costs while maintaining comparable accuracy to baselines. The ablation studies effectively demonstrate that these gains come from both the graph-based reasoning structure and the model allocation strategy, showing the value of each architectural choice.

3. Resource-efficient implementation: The plug-and-play adapter design is particularly practical, adding only 13M parameters (approximately 1/800th of the base model size) while enabling effective task allocation. This lightweight approach to improving edge-device capabilities, combined with the preservation of base model parameters, makes the solution highly deployable in resource-constrained environments.

CONS:
1. Insufficient evaluation of core architectural components: The paper lacks rigorous individual evaluation of its key components - task decomposition and dependency graph construction. There's no ablation study showing how different decomposition strategies affect performance, no comparison with other decomposition methods, and no metrics for measuring decomposition quality (e.g., granularity, completeness, independence). Similarly, while the dependency graph is central to the framework, there's no analysis of its construction accuracy or how different graph structures impact overall performance. This makes it difficult to validate the effectiveness of these fundamental components beyond their contribution to end-to-end performance.

2. Limited exploration of generalization and adaptability: Several critical aspects of the framework's generalization remain unexplored. First, the paper doesn't clarify whether a single task allocator adapter can handle all task types or if separate adapters are needed for different task domains (e.g., math vs. logic vs. web browsing). Second, while they evaluate on 7 benchmarks, most are focused on structured reasoning tasks, with limited testing on more open-ended or interactive scenarios. Additionally, there's no discussion of how the framework handles dynamic task requirements or adapts to new task types not seen during training. These limitations raise questions about the framework's broader applicability and deployment flexibility.

Overall, the Division-of-Thoughts paper presents a technically innovative and practically effective framework for edge-cloud LLM collaboration, demonstrating significant reductions in computational costs while maintaining reasoning accuracy through its well-designed three-component architecture and resource-efficient adapter implementation. However, the work has notable limitations in its evaluation methodology, particularly in validating individual components like task decomposition and dependency graph construction, and leaves important questions about cross-domain generalization and adapter specificity unanswered.

**Questions:**

Minor writing fixes:
1. Figure 3 caption has the word `[figure]` which can probably be removed.
2. Casing issues for the first word in line 150.

Some Technical questions:
1. Its unclear from the implementation details about the LLM used for task decomposition and scheduling strategies. Is the DoT framework expected to use cloud LLMs for these components?
2. As also mentioned in cons, how do the task decomposition and scheduler components generalize across various tasks?  Also, how do they scale with number of sub-tasks?
3. Usage of plug and play adapter for different tasks is unclear, are we training a single adapter for all the tasks evaluated?

**Reviewer Confidence:**

4: The reviewer is certain that the evaluation is correct and very familiar with the relevant literature

**Scope:**

4: The work is relevant to the Web and to the track, and is of broad interest to the community